# Pre-Treat Xenogenic Collagenous Blocks of Bone Substitutes with Saline Facilitate Their Manipulation and Guarantee High Bone Regeneration Rates, Qualitatively and Quantitatively

**DOI:** 10.3390/biomedicines9030308

**Published:** 2021-03-17

**Authors:** Stephane Durual, Leandra Schaub, Mustapha Mekki, Daniel Manoil, Carla P. Martinelli-Kläy, Irena Sailer, Susanne S. Scherrer, Laurine Marger

**Affiliations:** 1Biomaterials Laboratory, Division of Fixed Prosthodontics and Biomaterials, University of Geneva, University Clinics of Dental Medicine, 1, rue Michel Servet, 1204 Geneva, Switzerland; leandra.schaub@etu.unige.ch (L.S.); mustapha.mekki@unige.ch (M.M.); susanne.scherrer@unige.ch (S.S.S.); laurine.marger@unige.ch (L.M.); 2Division of Oral Diseases, Department of Dental Medicine, Karolinska Institutet, 17177 Stockholm, Sweden; daniel.manoil@ki.se; 3Laboratory of Oral & Maxillofacial Pathology, Division of Oral and Maxillofacial Surgery (HUG), Department of Surgery, University of Geneva, University Clinics of Dental Medicine, 1, rue Michel Servet, 1204 Geneva, Switzerland; CarlaPatricia.Martinelli-Klay@hcuge.ch; 4Division of Fixed Prosthodontics and Biomaterials, University of Geneva, University Clinics of Dental Medicine, 1, rue Michel Servet, 1204 Geneva, Switzerland; irena.sailer@unige.ch

**Keywords:** bone substitute, animal experiments, bone regeneration

## Abstract

Deproteinized bovine bone mineral particles embedded in collagen (DBBM-C) are widely used for bone regenerations with excellent, albeit sometimes variable clinical outcomes. Clinicians usually prepare DBBM-C by mixing with blood. Replacing blood by saline represents an alternative. We investigated if saline treatment could improve DBBM-C i. handling in vitro and ii. biological performances in a rabbit calvarial model. In vitro, DBBM-C blocks soaked in saline or blood were submitted to compression tests. In vivo, four poly ether ether ketone (PEEK)cylinders were placed on 16 rabbit skulls, filled with DBBM-C soaked in blood or saline for 2–4–8–12 weeks before histomorphometry. DBBM-C blocks were fully hydrated after 30 s in saline when 120 s in blood could not hydrate blocks core. Stiffness gradually decreased 2.5-fold after blood soaking whereas a six-fold decrease was measured after 30 s in saline. In vivo, saline treatment allowed 50% more bone regeneration during the first month when compared to blood soaking. This difference was then no longer visible. New bone morphology and maturity were equivalent in both conditions. DBBM-C saline-soaking facilitated its handling and accelerated bone regeneration of highly qualitative tissues when compared to blood treatment. Saline pretreatment thus may increase the clinical predictability of bone augmentation procedures.

## 1. Introduction

Contemporary bone regeneration is inseparable from matrices maintaining or defining a harmonious anatomical volume and capable of driving the target cells. Auto-, allo-, xeno-genic or alloplastic grafts act as a scaffold that will help to support new bone growth.

Within that respect, osseoconduction (guiding the regenerative cells), osseoinduction (inducing bone cells differentiation) as well as a controllable resorption rate are critical properties influencing new bone formation process [1]. For effective support of new bone formation, scaffolds must be developed with an appropriate chemical composition (e.g., biocompatible polymers, bioceramics) and structural specifications among which an interconnected macroporosity (ca. 60% range) and a surface micro roughness of ca. 2–3 µm. Those parameters are essential to allow for i. the adsorption of macromolecules and growth factors from the local environment, ii. the sprouting and the development of the vasculature and iii. the bone cells migration [2,3,4,5].

Bone scaffolds come in many different forms like blocks and particles (CaP bioceramics either synthetic, xenogenic, allogenic and autogenic), hydrogels, macromolecular foams/membranes/nets (e.g., gelatin, cross-linked collagens, Fibrin), phosphocalcic cements and various combinatory forms e.g., putties (macromolecular pastes and CaP particles) or soft blocks (CaP particles embedded in macromolecular foams) [6,7,8].

Most of the bone augmentations, small reconstructions and regenerations are performed by using particles covered by resorbable membranes that stabilize the construct and avoid loss of particulate material. Soft blocks made of macromolecular matrix embedding the particles are also widely used for bone augmentations of smaller size such as e.g., socket preservations. These hybrid blocks are generally composed of a matrix made of xenogenic collagen (bovine, equine or porcine) in which the particles are evenly distributed [9]. Particles handling and morphological adaptation to the defect are largely improved [10,11] while the bone regeneration process is supported at high and predictable levels [12,13].

Biological mechanisms of regeneration start from the adsorption of proteins, macromolecules and growth factors from the local environment on the injured tissue surfaces to serve as anchor layer for blood clot formation including activation of platelets and recruitment of inflammatory cells. The resulting granulation tissue will next initiate the migration of regenerative and vascular cells on site [14]. The same sequence will take part on and within the bone scaffold. Clinically, it is generally recommended to soak the scaffolds in the blood from the surgical site precisely to promote these mechanisms. Consequently, scaffolds range of soaking capacity for blood and sequential blood clot formation is of great importance as it may condition a proper initiation of the regeneration process. Blood absorption capacity is defined via several factors, amongst those, hydrophilicity has to be mentioned in particular.

Particulate bone scaffolds are not highly hydrophilic, either embedded in a collagen matrix or not [9,15]. Handling, malleability, water adsorption and soluble factors adsorption are necessarily impacted. As an example, deproteinized bovine bone mineral particles embedded in porcine collagen (DBBM-C), despite providing excellent clinical results [16,17,18,19,20], do not always allow a proper control of the bone regeneration [21,22,23], the composition and-or the level of compaction (per se the architecture) being potentially implied [21,22]. As an alternative to blood impregnation, clinicians are proposed by manufacturers to pre-soak these types of grafts in saline (NaCl 0.9%), the blood being adsorbed once the pre-soaked grafts are implanted.

Moreover, it was shown in a recent preclinical work on rat extraction sockets filled with a synthetic biphasic bioceramic bone substitute either hydrated in saline or blood before placement, that saline treatment significantly improves bone regeneration after 7- and 42-days regeneration [24].

Based on these findings, we hypothesized that the effect of pre-soaking in saline could also apply to hybrid blocks to

1-improve the handling and reproducibility of the manual application of the block material due to better malleability and thus compaction range,2-increase the quantity of new bone formed,3-increase the predictability of the clinical outcome due to homogeneous scaffold conditioning thanks to volumes of NaCl 0.9% easily controlled (when blood availability may be weak and critical on the surgical site.)

To the best of our knowledge, there is no study reported directly comparing effects of a DBBM-C saline pre-soaking on bone regeneration versus a pre-soaking with blood.

In this context, the objectives of that work were to evaluate the influence of a saline (NaCl 0.9%) pre-treatment (vs blood) of DBBM-C scaffolds on:-the compression forces as indicators for malleability of the blocks in an in vitro series,-the amount and kinetics of bone tissue formation from 2 to 12 weeks after implantation by using a rabbit calvarial model.

The study was designed to confirm or reject the null-hypothesis i.e., no difference between both types of pre-treatment procedure of DBBM-C.

## 2. Materials and Methods

### 2.1. In Vivo Experimental Design

In vivo testing was done according to the recently developed rabbit calvarial model [25]. Four individual cylinders were placed on rabbit skulls, on anatomical locations defined by the crossing of the median and coronal sutures (occipital left (OL) and right (PR), frontal left (FL) and right (FR)). The cylinders were filled with DBBM-C pre-treated with NaCl 0.9% (test group 1, n = 6 at each time point), DBBM-C pre-treated with blood from the ear vein (test group 2, n = 6 at each time point) and blood coagulum (blood from the ear vein that was left to coagulate before placement (control sham, n = 4 at each time point) (Table 1). Randomization was obtained by using the following distribution table:-Each condition was positioned at least once in cylinders OL, OR, FL, FR at each time point.-Each rabbit had to receive the three conditions at each time point.-Each control was to be placed on different animals in one of the four cylinders OL, OR, Fl and FR at each time point.

At 2, 4, 8 and 12 weeks, 4 animals were sacrificed, skulls were sectioned, cylinders were removed, and biopsies were processed for histology and histomorphometry (Figure 1A). Relative volumes of new bone and bone substitute were assessed.

The protocol herein described is compliant with the ARRIVE guidelines. It was approved by our local academic committee and the cantonal and federal veterinary agencies (authorizations n° GE/100/18, accepted on 9 July 2018).

### 2.2. Bone Substitutes

We used blocks of deproteinized bovine bone mineral particles embedded in porcine collagen (DBBM-C, Bio-Oss^®^ Collagen, Geistlich Pharma AG, Wolhusen, CH, Switzerland). These blocks are made of 90% deproteinized bovine bone mineral particles (0.25–1 mm in diameter) embedded in porcine collagen (10%). The total porosity reaches ca. 60% [9].

### 2.3. Cylinders and Screws

We designed cylinders with specific dimensions: inner diameter, 5 mm, outer diameter of 8 mm and a height of 5 mm. Lateral tabs were added to stabilize and fix the cylinders by screwing (Figure 1E). Finally, caps to close the cylinders hermetically (thickness of 1 mm) were also designed. Cylinders and caps were made of poly ether ether ketone (PEEK) (Boutyplast, Leyment, France).

We used micro screws (Global D, Brignais, France) made of CpTi gr5 to fix the cylinders (1.2 mm in diameter, 4 mm in length). PEEK cylinders, caps and screws were sterilized by autoclaving before surgery.

### 2.4. Animals

16 New Zealand white rabbits (male or female, 3 months, ca. 2.5 kg) were included in the study (UNIGE breeding, Arare, Switzerland). One week of acclimation was observed in our housing facility before surgery. A prophylactic antibiotic (Enrofloxacie Baytril 10%, 5–10 mg/kg po, Bayer, Leverkusen, Germany) was dispensed daily 2 h before surgery until the third day after surgery.

### 2.5. Surgical Procedure

The model was thorough described previously [25]. Briefly, animals underwent:Pre-anesthesia, im injection: ketamine (25 mg/kg, 50 mg/mL, 0.5 mL/kg, Pfizer, NY, USA) plus xylazin (3 mg/kg, 20 mg/mL, 0.15 mL/kg, Bayer).Anesthesia: propofol 2% iv (Braun, Sempach, Switzerland). Animals were then intubated and ventilated with sevoflurane 3% (Abbvie, Chicago, IL, USA) in pure oxygen. Analgesic solution of remifentanil (Bichsel, Unterseen, Switzerland) was continuously perfused iv (ear vein, 0.008–0.5 μm/kg/min, 5 g/mL).Surgery: skull was shaved, disinfected with povidone iodine solution 10% (Mundipharma, Frankfurt am Main, Germany) and locally anesthetized (lidocaine 2% sc, Sintetica, Mendrisio, Switzerland). A midsagittal incision was made through the skin and the periosteum which was gently elevated (Figure 1B). Four PEEK cylinders were screwed, five intramedullary holes were drilled under saline irrigation (0.8 mm in diameter, ca. 1 mm in depth) on the calvarium, within the perimeter of each cylinder, according to a precise template (Figure 1E). Cylinders were filled with bone substitutes materials and capped (Figure 1C). The surgical site was closed by using intermittent non-resorbable sutures (Prolene 4.0, Ethicon, Somerville, NJ, USA) (Figure 1D).

Quantities of bone materials to fill the cylinders were precisely calibrated and prepared in advance in a sterile way. Each cylinder was filled with a DBBM-C block of 74 mg, soaked and mixed (until complete impregnation) in NaCl 0.9% or fresh blood from the ear vein for 1 min precisely.

Post-surgical treatment: Analgesia was dispensed by sc injections of Buprenorphine hydrochloride (Reckitt Benckiser, Slough, UK) every 6 h for 3 days (0.02 mg/kg, 0.03 mg/mL, 0.67 mL/kg). Sutures were removed after ca. 10 days of healing.

### 2.6. Histological Preparation and Histomorphometric Analysis

At 2, 4, 8 and 12 weeks, four animals were sacrificed, skulls were sectioned, PEEK cylinders were removed and biopsies were fixed in formalin 4% (Sigma, St. Louis, MO, USA) for one month. Samples were then decalcified in EDTA (Osteosoft, Sigma) for three weeks and dehydrated before paraffin embedding. Samples were sectioned at three different levels (C1, C2, C3) according to a standardized and precise procedure (Figure 1F) planned to always evaluate regions of the same biological situation that is:-sagittal block section into two equal parts,-three sections with a range of 150 µm and a thickness of 5 µm.

Two histological staining were performed on each slice (Hematoxylin-Eosin and Masson-Goldner).

Slices were observed under a stereo video microscope (VHX-5000, Keyence). Particles and new bone tissue were analyzed using the Keyence analysis software. The structures of interest were delimited manually on the entire surface inside the PEEK cylinder, that is, from the external cortical limit of the bony bed to the internal limits of cylinder. The total new bone and bone substitute volumes were expressed as percentages of the total volume in the cylinder. For each sample, the total new bone was the mean value calculated from the 3 levels of cut C1–C3. Measurements were repeated two times by independent investigators.

### 2.7. Mechanical Testing

#### 2.7.1. Venous Blood Collection

Venous blood was collected from one healthy donor into tubes containing trisodium citrate 0.129 M (blood-citrate 9/1) and kept at 4 °C for a maximum of 1 day. Five minutes before mechanical testing, the blood was reconstituted with a solution of calcium chloride 1M (Applied chem, Darmstadt, Germany) to a final Ca concentration of 20 mM.

#### 2.7.2. Passive Hydration, Semi-Quantitative Colorimetric Evaluation

One ml of complete fresh blood, NaCl 0.9% or fetal calf serum colored by bromophenol blue were deposited on DBBM-C blocks (7 × 7 × 8 mm) for 30 and 120 s. Blocks hydration and impregnation were done passively, without mixing the blocks with the liquids. In a fourth set up, blocks were briefly soaked with NaCl 0.9% followed by 30 s exposure to fresh blood. Blocks were then cut into two equal parts and photographed instantly. Hydration was visually evaluated based on the photo documentation. Three separate experiments were realized in triplicate.

#### 2.7.3. Malleability: Compression Test

Malleability is defined as the ability to deform and change shape under compressive stress and is measured by the ability of a device to withstand pressure. One ml of complete fresh blood or NaCl 0.9% were deposited on DBBM-C blocks (7 × 7 × 8 mm) for 30, 60 and 120 s. Blocks’ hydration and impregnation were done passively, without mixing the blocks with the liquids. In a third set up, blocks were briefly soaked with NaCl 0.9% followed by 30 s exposure to fresh blood. Blocks were placed on the compression plate of a Universal Testing Machine AGX plus (Shimadzu Corporation, Kyoto, Japan) equipped with a 500 N force sensor, before they were submitted to a continuous vertical compression (1.0 mm/min, 0–500 N, sampling frequency 2 Hz). Compression curves were drawn, and initial stiffness was calculated from the slopes between 1–15 N of these curves (1–15 N represents the force range applied by manual handling with a surgical tool. This reference range is based on the measurements of the force applied with a spatula on the force sensor that was used for the compression tests (3 different operators, data not shown). Three separate experiments were realized in triplicate.

### 2.8. Statistical Analysis

New bone volume data were checked for normal distribution and equivalence of variances. Unpaired *t*-tests were used to compare NaCl 0.9% and blood series at 2, 4, 8 and 12 weeks. The null hypothesis was rejected at *p* < 0.05.

Stiffness data were checked for normal distribution and variances analysis, followed by a Bonferroni post-hoc test. The null hypothesis was rejected at *p* < 0.05.

## 3. Results

### 3.1. Mechanical Testing: DBBM-C Malleability

Prior to in vivo analysis, mechanical testing assays were performed on DBBM-C scaffolds either soaked with blood or NaCl 0.9%.

In a first set of experiments, the passive hydration of DBBM-C was evaluated. After 30 s of contact with blood, a thin peripheral layer of ca. 1 mm was hydrated whilst the block core was still dry (Figure 2(Aa)). After 120 s, blood failed to impregnate the whole block and penetrated roughly 2 mm of the block periphery (Figure 2(Ab)). In contrast, a 30 s treatment with NaCl 0.9% (Figure 2(Ac,d)) or serum (Figure 2(Ae,f)) was sufficient to hydrate the whole block. Finally, an instant NaCl 0.9% soaking allowed for an entire blood penetration within the blocks after 30 s contact (Figure 2(Ag)).

In a second set of experiments, we evaluated blocks stiffness. Compression was first measured up to a force of 500 N. Initial stiffness was calculated from the compression curves slopes between 1–15 N (i.e., the force range applied by manual handling with a surgical tool) and was assimilated to malleability.

Scaffolds soaked in NaCl 0.9%, blood or NaCl 0.9% + blood were submitted to compression forces from 0–500 N. As shown by compression curves (Figure 2B), soaking with blood for 30 or 120 s lightly increased blocks compressibility when compared to a dry block. For example, to reach a compression of 3 mm, ca. 150 N were necessary for a dry block. After 30 and 120 s in blood, ca. 110 and 125 N were needed, respectively. By comparison, compressibility was increased ca. 5-fold with a NaCl 0.9% treatment, blocks being soaked in saline for 30 s needing ca. 25 N for a compression of 3 mm.

Blocks initial stiffness (Figure 2C) was not modified after 30 s of soaking with blood when compared to the control dry block. From 30 to 120 s, blood soaking led to a gradual decrease of the stiffness that was ca. 2.5 times lower after 120 s. At contrary, the lowest stiffness of 10 Nmm^−1^ was reached after 30 s of soaking in NaCl 0.9%, that represented a 6-fold decrease with respect to the control.

### 3.2. Clinical Course and Macroscopical Evaluation at Time of Sacrifice

All animals survived the surgical procedure well, and we did not deplore any loss of rabbits. No clinical signs of inflammation at the site of the cylinder placement were observed throughout the entire in life observation period. At harvest of the cylinders no inflammation related changes were visually detectable.

In case of the sham sites, only limited amount of new tissue was formed in close contact to the pristine bone at all time points. In case of DBBM-C new tissue had formed that increased with time until finally at 12 weeks the whole cylinder volume was filled with bone like appearing tissue. No difference between DBBM-C NaCl 0.9% and DBBM-C blood was evident.

### 3.3. Histologic Evaluation

We could not detect qualitative differences from a histological perspective between the cylinders filled with DBBM-C scaffolds pre-treated with blood or NaCl 0.9%. Thus, the herein described observations apply to both conditions.

At all time-points the particles were evenly distributed within the cylinders (Figure 3A,B and Figure 4A,B). Three distinct layers of tissue were detectable: (i.) granulation tissue, (ii.) an osteogenesis zone characterized by the presence of osteoid and (iii.) a bone remodeling zone in close contact to the pristine calvarial bone (Figure 3A,B). The three distinct tissue type components varied in location and ratio depending on the time points under investigation. As such the granulation tissue zone migrated from the bottom of the defect to the top followed by the osteogenesis zone that decreased gradually to the benefit of the remodeling zone. All three different section levels per sample displayed the same distribution of the three zones.

Granulation tissue: The granulation tissue was richly vascularized, and of mainly mononuclear character also comprising some polymorphonuclear cells (Figure 3A–C). The tissue embedded the DBBM particles rather homogeneously, no accumulation of inflammatory cells in close vicinity of the granules was obvious. This zone had “migrated” from the bone bed up to the top of the cylinder at 12 weeks.

Osteogenesis zone: Below the granulation tissue zone the particles were embedded in fibrous appearing highly vascularized tissue (Figure 3A,B). Higher magnification of this zone showed that the particles were surrounded by rows of osteoblasts or deposits of fringes of osteoid or woven bone isles also lined by rows of osteoblasts (Figure 4B,C). Presence of osteoclast was also observed at the mineral particles surface, followed by rows of osteoblast, thus depicting basic multicellular units (Figure 4C). With time the osteogenesis zone moved upwards without change of its composition.

Remodeling zone: New bone tissue in form of trabeculae was observed within the first millimeter above the bone bed, sprouting from the transcortical holes and the bone bed (Figure 4A,B,D) already at two weeks. Between the aggregates of bone and particles a vessel rich loose connective tissue was observed (Figure 4D). The amount of bone increased with time to such an extent that at 12 weeks the cylinders were largely filled with new bone surrounding the particles from the bone bed up to the top (Figure 4E). Signs of remodeling were clearly observed as shown by the delimitation between the bone bed and the new bone within the cylinder that was no longer visible (Figure 4E). The presence of lamellar bone demonstrated that the tissue had gained in maturity, as well as the large inclusions of bone marrow within the spaces delimited by the particles and the bone (Figure 4F). The particles were completely integrated within the new bone tissue and did not show any sign of resorption (Figure 4F).

### 3.4. Histomorphometric Evaluation

Granulation tissue, new bone tissue, and particles were addressed in these analyses.

As shown by histological analyses, the particles were evenly distributed, not resorbed and filled 40% of the cylinders.

The granulation tissue (GT) gradually migrated from the bone bed up to 4 mm in DBBM-C scaffolds pretreated with blood or NaCl 0.9%. We observed a tendency for the GT to migrate more promptly during the first month in NaCl 0.9%-DBBM-C blocks when compared to blood pretreated blocks (Figure 3D). These differences were not significant but were correlated to the kinetics of new bone formation. Indeed, the new bone tissue increased in total volume gradually from 2–12 weeks, independently from the DBBM-C scaffolds pre-treatment. However, within the first month, the new bone increased by ca. 50% when the scaffolds were treated with NaCl 0.9% as compared to blood (3.4 ± 0.7% vs. 2 ± 0.5% of the total area, respectively, *p* = 0.01). This difference was no longer visible at 8 and 12 weeks, the maximal amount of new bone within the cylinders reaching 8.9 ± 1.5% with a NaCl 0.9% pre-treatment and 8.3 ± 0.9% with a blood pre-treatment (Figure 5A, *p* = 0.5). The statistical repartition of the values obtained on both series was homogeneous at each time point, mean and medians being merged most of the time (Figure 5B). Finally, we did not observe significant differences between the three different levels of cut in terms of new bone volume (data not shown).

Regarding the vertical bone growth (Figure 6), the new bone grew in close contact with the bone bed at two weeks. At 4 weeks, bone had grown in the upper compartments of the cylinder reaching 2–3 mm in height with a NaCl 0.9% pre-treatment (62% of the new bone between 0–1 mm, 30% between 1–2 mm and 7% above 2 mm) and not exceeding 1–2 mm with blood (80% of the new bone between 0–1 mm, 20% between 1–2 mm). These differences between the two scaffolds pre-treatments remained visible but not significant at 8 and 12 weeks. A new vertical layer was reached by the new bone at each time point, i.e., above 3 mm at 8 weeks and above 4 mm at 12 weeks.

## 4. Discussion

The present work aimed at investigating the effects of a saline pre-soaking of deproteinized bovine bone mineral particles embedded in porcine collagen (DBBM-C) on the rate and quality of bone regeneration in a rabbit calvarial model over a period of 12 weeks. DBBM-C handling was also evaluated in series of in vitro mechanical testing. Malleability and hydrophilicity of DBBM-C blocks were largely improved by the NaCl 0.9% pre-treatment as compared to blood in vitro. When compared to a blood pre-treatment, new bone growth was increased during the first month of regeneration by ca. 50%. The quality and maturity of the bone tissue was equivalent at completion in both conditions.

DBBM-C bone substitutes are daily used clinically and proved satisfactory in terms of adaptation to the defect, stability over time and support for bone regeneration. However, it seems that bone regeneration is not always predictable, probably due to variable compaction of the blocks. Most of the surgeons prepare DBBM-C by soaking in blood from the surgical site before placement. A saline pre-treatment may be an interesting alternative, either biologically, clinically or practically. To evaluate this alternative protocol, we developed a study divided into two parts, one focused on in vitro mechanical testing of DBBM-C blocks pre-treated with saline, the second being devoted to the biological effects of DBBM-C blocks following the same treatment before implantation in vivo. Blood pre-soaking was used as a reference in both protocols.

Assessing manipulation gains thanks to saline pre-treatment was the aim of the first part of that study. By using a standardized method in vitro, we observed a malleability (that we extrapolated from the stiffness) after 30 s soaking that was six times higher with saline than with blood. The stiffness was measured between 1–15 N, a force range corresponding to the force applied by hand with a spatula to mix the block with liquids (either blood or saline). This range was chosen after we measured the force applied with this spatula on the force sensor that was used for the compression tests (three different operators, data not shown). This is in line with forces applied during bone wax manipulation–i.e., a material that is close to hydrated DBBM-C blocks in texture–and that reach ca. 4 N [26].

We assume that these advantageous mechanical effects observed in these in vitro series come from a wider wettability of saline-DBBM-C blocks as shown by passive hydration assays. The blocks showed almost the same wettability while they were soaked in serum or saline. However, when using complete blood, we observed a lack of core block hydration that remained dry even after 2 min soaking. This could be explained by a pore blockage by erythrocytes that pile up and form a crown around the block core thus stopping liquid influx. However, these findings cannot explain the effects we observed in vivo on bone formation, the blocks being mixed thoroughly in presence of blood or saline, guarantying therefore a complete material hydration as well as a homogeneous red cells distribution, as discussed later.

In order to complete the second part of that study, we used a calvarial model of bone regeneration. Basically, it consists in the fixation of individual reactors in which bone regeneration may occur towards the vertical direction. The extend of bone formation depends on the provision of scaffolds since blood coagulum alone is not sufficient to support complete bone formation in the vertical dimension. This makes it the ideal experimental tool to evaluate e.g., new bone substitutes and their manipulation, osteogenic factors as well as biological mechanisms like neovascularization and osteogenesis. Biologically, the principle is to grow new bone tissue above a cortical plateau, a situation that can be compared to a class 4 defect in the jaw [27].

Depending on the animal used (e.g., sheep, pigs, rabbits, rats, etc.), 2–8 reactors may be fixed on the skull. The rabbit appears as a rationale choice, first because of quite similitudes with human bone metabolism and structures [28] and secondly for practical reasons as four reactors may be placed simultaneously [25,29]. As a proof of the relevancy of this animal model for bone regeneration, about 80% of the references based on the calvarial model used rabbits.

Independently from the DBBM-C pre-treatment, we measured with this model relative new bone tissues volumes of ca. 10% after 12 weeks. These rates are weak when compared to sheep for example where relative volumes of new bone may reach 40% over the same time course by using a calvarial model [30,31]. However, they are in line with data obtained with almost the same model in the rabbit in which DBBM-C supported ca. 5% of new bone growth after 12 weeks implantation [32], bovine bone particles 11% after four weeks [33]. It is important to be noted also that the bone tissue that we observed after 12 weeks was highly mature, showing lamellar structures and bone marrow inclusions, even at the top of the biopsies. Finally, we must point out that the in vivo model was aimed at assessing biological performances only. DBBM-C blocks were indeed soaked in blood and saline until complete impregnation and maximal malleability were reached, so that handling, level of compaction and manipulation during placement could not be evaluated. We have besides measured relative volumes of particles and their repartition that were equivalent for both conditions.

Despite that the quality of the bone tissue was equivalent when DBBM-C blocks were soaked in blood or saline before placement, kinetics of bone formation were widely different. In effect, during the first month implantation, we observed a 50% increase of bone formation when the blocks were pre-treated in saline rather than in blood. Vertical conduction was also improved as demonstrated by higher levels in height reached by the new bone tissue grown in saline-DBBM-C blocks.

Since a saline pre-treatment allows the development of healthy and mature bone tissue with improved bone growth kinetics during the first month when compared to blood pre-treatment, this alterative appears as a method of choice. Especially since saline is inexpensive, non-limiting and chemically well-defined when blood from the surgical site may be poorly abundant, clotted and biochemically highly variable. Thus, a calibrated procedure with excellent previsions of clinical outcomes may be easily defined based on these data.

The observed difference with regards to handling, blood interfusion and accelerated bone formation whilst providing the same bone quality warrants closer in-depth discussion. Three hypotheses are worthwhile to mention:i.Pretreatment with saline leads to better handling of DBBM-C i.e., increased compaction efficiency. The better the compaction the higher the contact area of the DBBM-C material to the surface of the bone and the better the conditions for osseoconduction. The easier the application the better the predictability of the clinical outcome.ii.Saline respectively serum pre-treatment data clearly indicate a high level of wettability of the DBBM-C blocks. Therefore, serum components as well as bioactive factors from the local environment in vivo must be absorbed to the DBBM-C being pre-treated with blood or saline. Differences in quantity and composition of absorbed serum or local factors may be responsible for the observed acceleration and should be thoroughly investigated.iii.The finding that erythrocytes interfuse the DBBM-C block completely only after saline treatment in vitro argues for a “blockage” of the erythrocyte in case of using whole blood. The fact that the serum alone penetrates the block completely also argues in that way. However, we have seen that blocks were mixed thoroughly with blood before placement in vivo so that they were fully hydrated and colonized homogeneously by red blood cells. Yet, we observed a delay in the vertical migration of the granulation tissue in blood-soaked blocks as compared to saline, correlated to a delay in bone formation. The same observation i.e., more bone formation in the same time-period in case of saline pre-treatment of biphasic bone substitute granules was reported by Santos et al. [24]. Thus, the observed phenomenon seems to be a general one and cannot be attributed to the composition and structure of DBBM-C alone. We believe that the reason for ours and Santos’ et al. findings may be the influence of saline on blood clotting i.e., fibrinogenesis. In the presence of saline, the activation of fibrinogen should be postponed and the evolving fibrin network less dense simply because of a reduced concentration of fibrinogen and its activators in situ. A postponed fibrinogenesis may increase the distance that erythrocytes may infuse before being trapped in the blood clot. A looser fibrin network will lead to a less rigid final product and thus better malleability and compaction. Moreover, such a loose fibrin network will allow for more rapid cell invasion in particular of polymorphonuclear cells [34] being the forefront of the healing cascade and thus accelerate the healing kinetics. Because collagen is a known activator of fibrinogenesis we predict that the effect of saline pre-treatment on bone formation kinetics will be more pronounced in case of bone substitute materials comprising particles only.

In conclusion, we identified a rapid, simple and efficient procedure of pre-treatment for deproteinized bovine bone mineral particles embedded in porcine collagen. Saline soaking before placement of these blocks of bone substitutes guarantees highly qualitative bone tissues growths, accelerated kinetics of regeneration and facilitated handling. It may be a very interesting alternative to traditional blood pre-treatment and should allow for standardized clinical protocols, better control of i. blocks compaction and ii. new bone tissue quality.

## Figures and Tables

**Figure 1 biomedicines-09-00308-f001:**
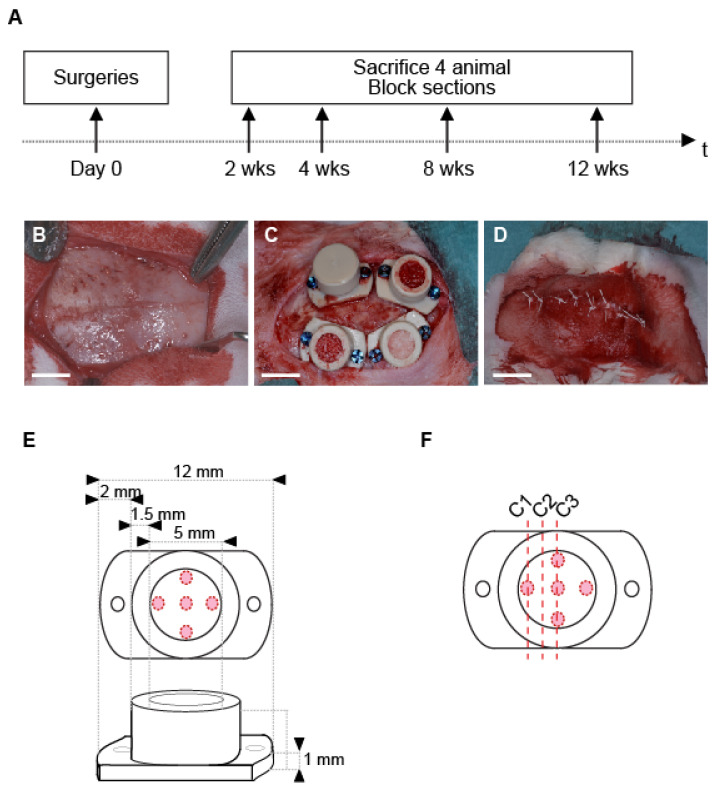
(**A**) Time frame of the study. (**B**–**D**) Main stages of the surgical procedure: (**B**) exposition of the calvarium, note the bone sutures that delineate 4 quarters; (**C**) placement and filling of the cylinders in the 4 quarters delimited by the bone sutures; (**D**) site suture; (white bars, 5mm); (**E**) Cylinder specification. Red circles mark the position of the intramedullary holes drilled on rabbit skulls; (**F**) Schema showing the three different levels of cut (C1–3) within the sample for histological and histomorphometric analyses.

**Figure 2 biomedicines-09-00308-f002:**
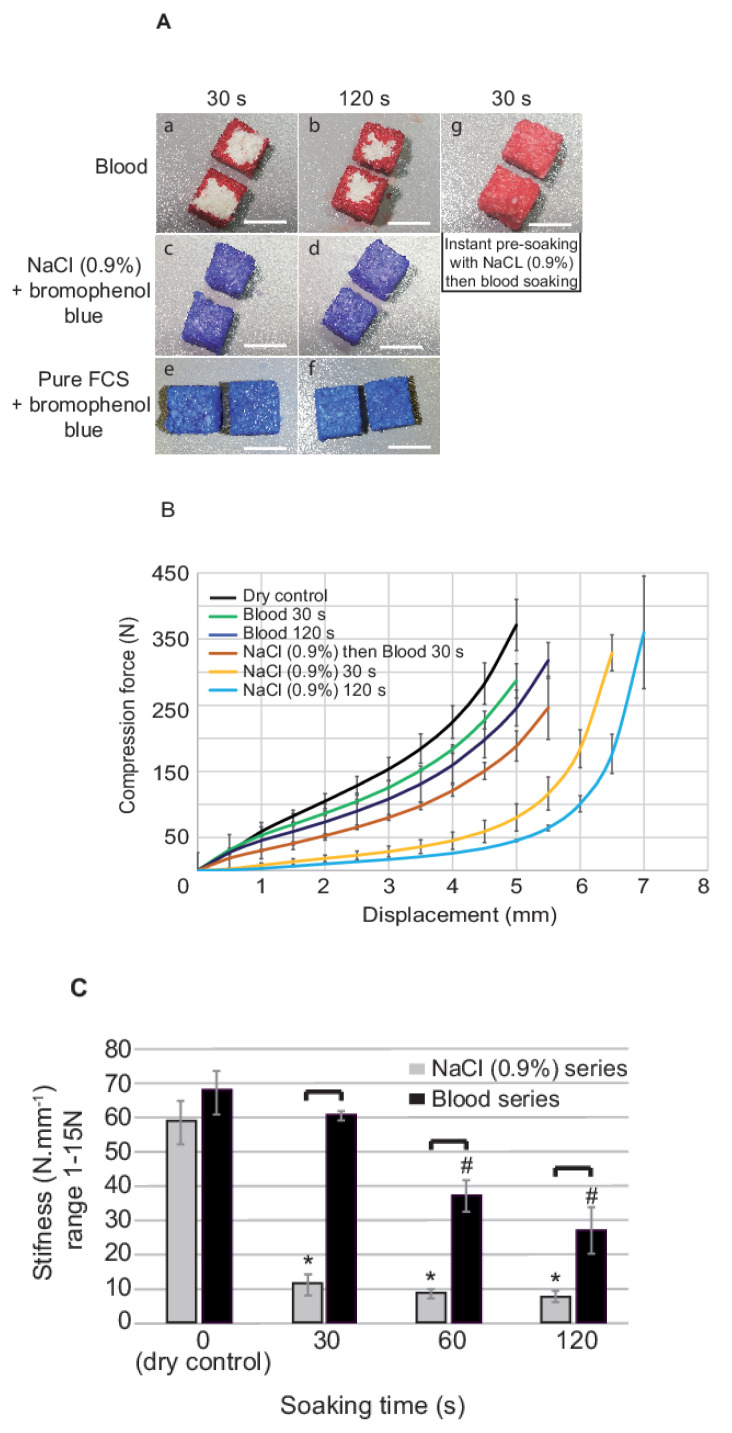
Mechanical testing and passive hydration assessment of the DBBM-C scaffolds. (**A**) DBBM-C scaffolds were soaked in fresh blood (**a**,**b**), NaCl 0.9%–bromophenol blue (**c**,**d**) and pure fetal calf serum (**e**,**f**) for 30 and 120 s. A last treatment consisted in an instant soaking with NaCl 0.9% followed by 30 s in fresh blood (**g**). Blocks were then cut into 2 equal parts and photographed instantly (Representative pictures, n = 3 per treatment; white bars, 5 mm). (**B**) Compression curves (load (0–500 N) vs. displacement (mm)) of the scaffolds treated with fresh blood or NaCl 0.9% for 30 and 120 s, instant NaCl 0.9% then fresh blood for 30 s (3 independent series (blood/NaCl 0.9% pre-treatment/NaCl 0.9% + blood), n = 3 per treatment at each time point; Reference: dry scaffold, results expressed as Mean ± SD). (**C**) Stiffness (slope between 1–15 N calculated from the compression curves) of the scaffolds with respect to the nature and time of pre-treatment (3 independent series (blood/NaCl 0.9% pre-treatment/NaCl 0.9% + blood), n = 3 per treatment at each time point; Reference: dry scaffold, results expressed as Mean ± SD, *: significantly different with respect to dry control NaCl series (*p* < 0.05); #: significantly different with respect to dry control Blood series (*p* < 0.05); Brackets: indicates two conditions significantly different (*p* < 0.05).

**Figure 3 biomedicines-09-00308-f003:**
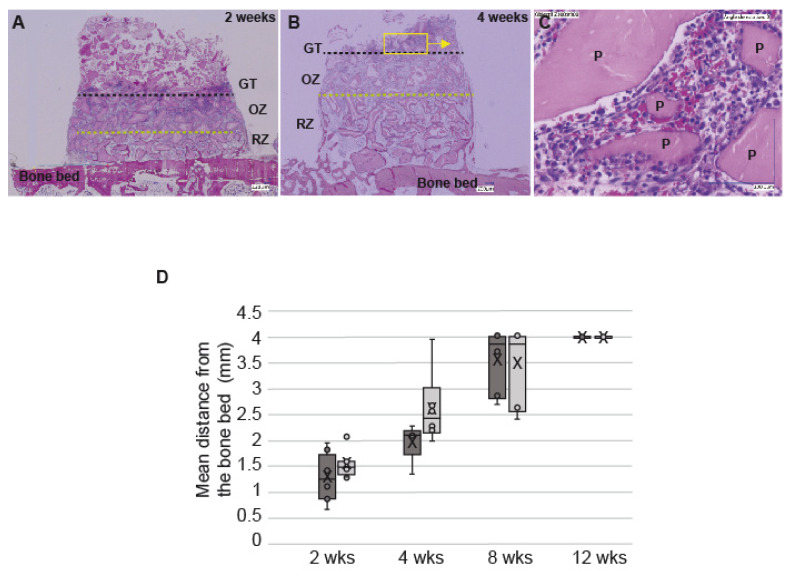
Migration of the granulation tissue. (**A**,**B**) representative histological slides showing the migration of the granulation tissue within cylinders filled with DBBM-C soaked in NaCl 0.9% at 2 weeks and 4 weeks (dotted black line, mean height of the granulation tissue (GT); dotted green line, inferior limit of the osteogenic zone (OZ) upper limit of the remodeling zone (RZ). Yellow box and arrow: magnification on the picture indicated by the arrow. (**C**) Higher magnification of the granulation tissue showing mononuclear cells surrounding the particles of DBBM-C. (**D**) Box plot showing the mean distance (mm) between the bone bed and the GT in the cylinders filled with DBBM-C scaffolds either pre-treated with blood (dark grey) or NaCl 0.9% (light grey) at 2, 4, 8 and 12 wks. Horizontal line: median; Boxes: 25–75%; ×: mean; Bars: range of non-outliers.

**Figure 4 biomedicines-09-00308-f004:**
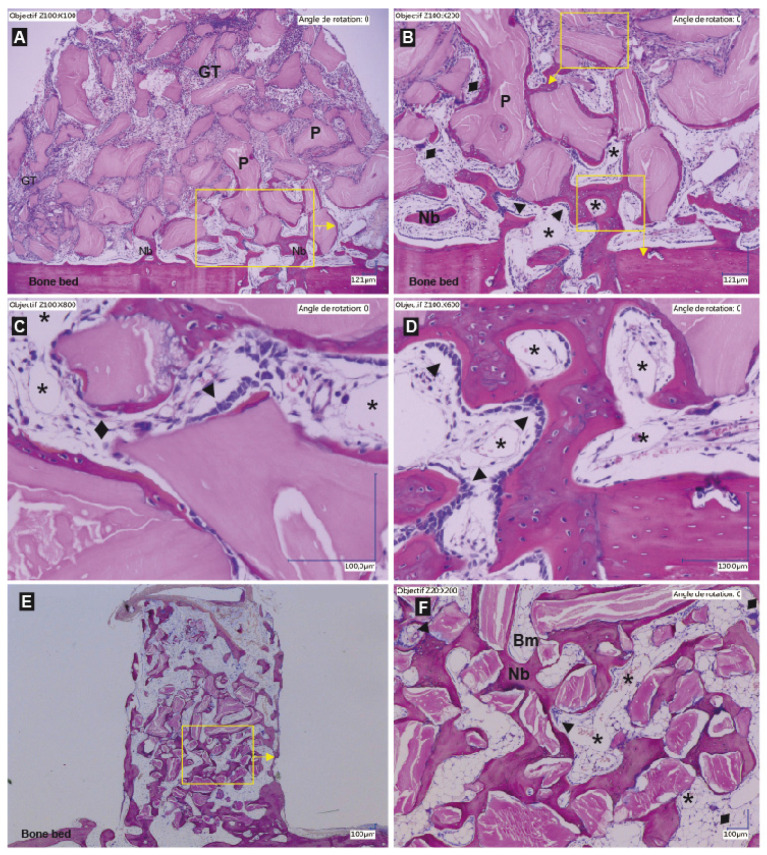
Representative histological slides showing the entire content of a cylinder filled with a DBBM-C scaffold pre-treated with NaCl 0.9% at 2 weeks (**A**) and 12 weeks (**E**) and higher magnifications ((**B**–**D**): 2 wks; (**F**): 12 weeks). Hematoxylin-eosin staining, particles (P) appears as pink, Bone and New bone (Nb) appear as purple. (GT: granulation tissue; Dark arrows ►: lines of osteoblast; Bm: bone marrow; *: capillaries; ♦: osteoclasts; Yellow boxes and arrows: magnifications on the pictures indicated by the arrows.) (**A**) At 2 weeks, the new bone sprouts from the bone bed and the transcortical holes. Osteoid and new bone tissue are observed in close vicinity to the bone bed. The granulation tissue has already migrated to the top of the scaffold. Below the GT is the osteogenic zone (**B**,**C**), a highly vascularized tissue where new bone is deposited at the mineral particle surface. Note the lines of osteoblasts that deposit layers of new bone around the particles (**B**,**C**). Some osteoclasts are also observed at the surface of the particles not yet osseointegrated, followed by osteoblast to form basic multicellular units (**C**). In close vicinity to the bone bed, the remodeling zone is found formed by new bone trabeculae and a vessel rich connective tissue (**D**). At 12 weeks, all the particles are osseointegrated and the bone has filled the whole cylinder (**E**). Note the signs of bone maturation with the presence of bone marrow foci in the bone lacunae as well as the presence of lamellar bone (**F**).

**Figure 5 biomedicines-09-00308-f005:**
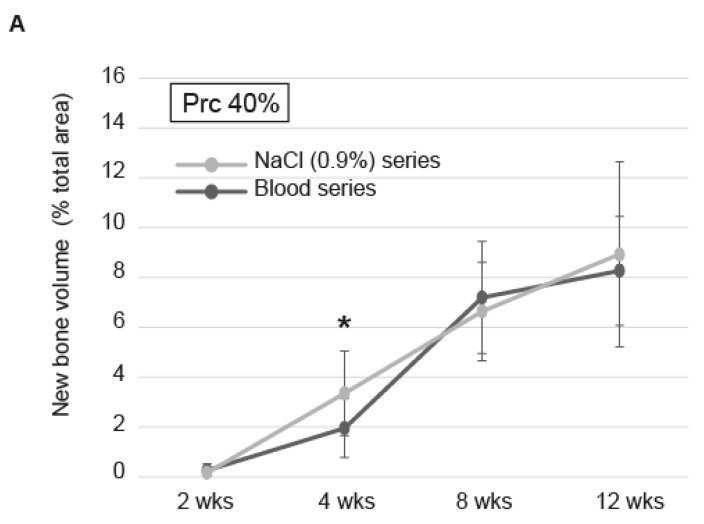
Histomorphometry (**A**) New bone volume in the cylinders filled with DBBM-C scaffolds either pre-treated with blood (dark grey) or NaCl 0.9% (light grey) at 2, 4, 8 and 12 wks. Data are expressed as mean ± SD, *: significantly different (*p* < 0.05) (n = 6 per treatment at each time point). (**B**) Box plots showing the new bone volume in the cylinders filled with DBBM-C scaffolds either pre-treated with blood or NaCl 0.9% at 2, 4, 8 and 12 wks. Horizontal line: median; Boxes: 25–75%; ×: mean; Bars: range of non-outliers. Brackets: significantly different (*p* < 0.05).

**Figure 6 biomedicines-09-00308-f006:**
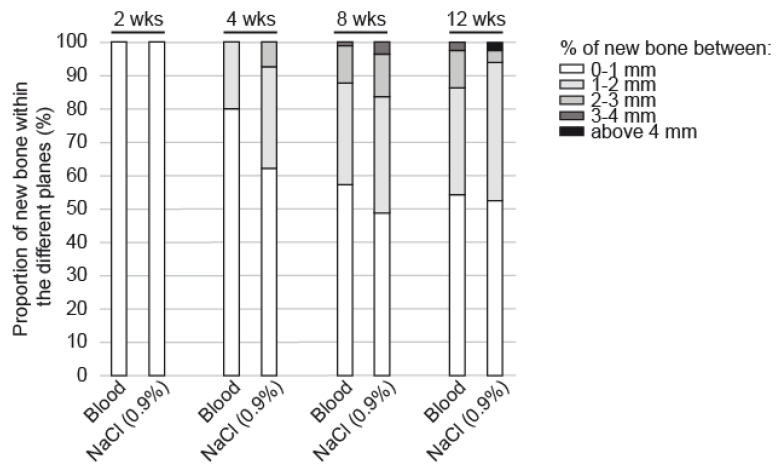
Proportion of new bone volume in the cylinders filled with DBBM-C scaffolds either pre-treated with blood or NaCl 0.9% segmented in 5 horizontal planes of 1 mm in height at 2, 4, 8 and 12 weeks.

**Table 1 biomedicines-09-00308-t001:** Specimen analyzed.

	Bone Regeneration Period				
Samples	2 wks	4 wks	8 wks	12 wks	Total
Sham blood coagulum	4	4	4	4	Placement 64 Analyzed 64
DBBM-C–NaCl 0.9%	6	6	6	6	
DBBM-C–Blood	6	6	6	6	

## Data Availability

The data that support the findings of this study are available from the corresponding author, upon reasonable request.

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
