# Peer review of "Pre-Treat Xenogenic Collagenous Blocks of Bone Substitutes with Saline Facilitate Their Manipulation and Guarantee High Bone Regeneration Rates, Qualitatively and Quantitatively"

_biomedicines, 2021, doi:10.3390/biomedicines9030308_

Round 1

Reviewer 1 Report

In general terms I understand that new methods for daily surgeries are needed and blood being a complex and indeterminate amalgam of factors, proteins or cells should be change to a more stable and determinate medium. Nevertheless, the proposal of incorporation of BMPs or other growth factors is more interesting and challenging that using NaCl 0.9%. No biological response is expected from the use of saline solution. General comments Figure 2. B improve the quality of the graphic, numbers and captions are too small, indicate the SD of each curve as it has n=3. Figure 2.C improve the graphic quality (Origin, Prism or similar software is recommended) indicate the Mean ± SD. The statistical analysis of this graphic is hard to interpret, the use of a or b is rare. Figure 5. Include mean± SD not SEM. All graphics should be improved. Discussion “Despite that the quality of the bone tissue was equivalent when DBBM-C blocks were 458 soaked in blood or saline before placement, kinetics of bone formation were widely dif- 459 ferent. In effect, during the first month implantation, we observed a 50% increase of bone 460 formation when the blocks were pre-treated in saline rather than in blood. Vertical con- 461 duction was also improved as demonstrated by higher levels in height reached by the new 462 bone tissue grown in saline-DBBM-C blocks.” Why the kinetics change? How the growth factors, proteins and cells present in blood can affect the final implant performance? Why the authors perform a mechanical evaluation of “DBBM-C scaffolds. (A) 267 DBBM-C scaffolds were soaked in fresh blood (a,b), NaCl 0.9% - bromophenol blue (c,d) and pure 268 fetal calf serum (e,f) for 30 and 120 sec.” but the they only use NaCl 0.9% or blood soaked in animal studies? The final outcome of the implants does not seem to change in final terms and the use of NaCl 0.9% seems to induce a more heterogeneous response relaying on the final capacity of each animal to regenerate the area. Did you compare the same animal with the different implants in order to consider the same regeneration ability? No blood was available in the area of implantation?

Reviewer 2 Report

In the current manuscript Durual et al. demonstrate the efficacy of saline pretreatment of deproteinized bovine bone mineral particles embedded in porcine collagen. They demonstrate the ability of saline to hydrate these blocks at a higher rate and extent than blood. They further demonstrate increased malleability and decreased stiffness in the saline treated group compared to both control and blood treated groups. interestingly although trends to enhanced bone formation were observed in vivo no significant differences were reported. all in all this is a well thought out well executed and well presented study. This reviewer did however have 2 minor comments:

1- while the saline did improve some of the mechanical characteristics of the DBBM blocks in vitro, in vivo the were scant differences. as such this reviewer feels rewording of the title is appropriate.

2- the data presented in figure 2 B would be greatly strengthened by including the statistical results for this experiment. 

Round 2

Reviewer 1 Report

The paper can be accepted in the present form.